# Effect of the Addition of Humic Substances on Morphometric Analysis and Number of Goblet Cells in the Intestinal Mucosa of Broiler Chickens

**DOI:** 10.3390/ani13020212

**Published:** 2023-01-06

**Authors:** Yair Román López-García, Sergio Gómez-Rosales, María de Lourdes Angeles, Héctor Jiménez-Severiano, Rubén Merino-Guzman, Guillermo Téllez-Isaias

**Affiliations:** 1Posgrado en Ciencias de la Producción y de la Salud Animal, Facultad de Estudios Superiores Cuautitlán, Universidad Nacional Autónoma de México, Km 1 carretera a Colón, Queretaro 76280, Mexico; 2Centro Nacional de Investigación Disciplinaria en Fisiología y Mejoramiento Animal, INIFAP, Km 1 carretera a Colón, Queretaro 76280, Mexico; 3Departamento de Medicina y Zootecnia de Aves, Facultad de Medicina Veterinaria y Zootecnia, Universidad Nacional Autónoma de México, Ciudad de Mexico 04510, Mexico; 4Department of Poultry Science, University of Arkansas, Fayetteville, AR 72701, USA

**Keywords:** broiler chickens, humic substances, intestinal villi, goblet cells

## Abstract

**Simple Summary:**

During recent decades, researchers have investigated humic substances (HS) as a potential replacement for, or reduction in the use of, growth-promoting antibiotics (GPA) in chicken diets. Several mechanisms of action have been proposed to explain the improvements in productive and health responses observed in broiler chicks given HS, such as stabilizers of the intestinal flora, effects on the digestive system similar to acidifiers, and the creation of protective layers on the epithelial mucous membrane of the digestive tract, but none of these theories have been demonstrated. Whether HS affects the histology and number of goblet cells (GC) in the intestine of broilers kept under stable digestive conditions and under abrupt changes in diet was studied in this research. From 14 to 38 days, HS-fed broilers had a similar FCR to AGP-fed broilers, but there was no clear trend in the effect of HS on the morphometry of villi in broilers raised under stable digestive conditions and following abrupt dietary changes. Under stable digestive conditions and after experiencing sudden dietary changes, the number of GC in the jejunum of HS-fed broilers behaves closely to that of AGP-fed broilers. These findings may aid in explaining the mechanism of action of HS.

**Abstract:**

The mechanisms of action of humic substances (HS) as growth promoters in poultry are unknown. In this study, the productive performance, histology, and number of goblet cells (GC) in the intestinal villi of broilers under steady-state digestive conditions and under abrupt changes in diet with the addition of HS was evaluated. Broilers housed individually were offered three treatments from 14–28 days: 1 = diet with white corn/soybean meal, without growth promoter antibiotics (nonGPA); 2 = with GPA (GPA); and 3 = with 0.3% HS. At day 28, two diets were suddenly introduced: (A) white corn/soybean meal plus dried distillers’ grains with solubles (DDGS); and (B) white/blue corn/soybean meal/DDGS, keeping the three original treatments. Diets A and B were also exchanged on day 37. FCR was lower with GPA and HS compared to nonGPA from 14–38 days (*p* < 0.05); at day 28, under steady-state digestive conditions, HS had a similar effect to GPA on the histology and GC number in the jejunum villi. The number of GC in the jejunum of HS-fed broilers on days 29 and 38, after diet changes, behaves similarly to that of AGP-fed broilers (*p* > 0.05). HS appears to strengthen the mucosal protection of the epithelium of the intestine.

## 1. Introduction

Researchers have investigated humic substances (HS) as a potential replacement for, or reduction in, the use of growth-promoting antibiotics (GPA) in chicken diets over the last two decades due to their ability to improve animal performance and health [1,2,3]. Several mechanisms of action have been proposed to explain the improvements in productive responses detected in broiler chicks given HS as a supplement, but none have been confirmed to date [4]. It has been proposed that HS stabilizes the intestinal flora and improves nutrient utilization from feeds, resulting in improved growth performance without increasing the amount of given feed [1]. Another theory proposes that HS have an effect on the digestive system that is similar to acidifiers, lowering the pH and acting as antimicrobials, thereby improving bird intestinal health and productivity [5,6,7]. Additionally, it has been suggested that HS have the capacity to form protective layers on the epithelial mucous membrane of the digestive tract, preventing the colonization of pathogenic bacteria or the absorption of toxins produced by bacteria because of their colloidal properties and high capacity to form aggregates within solutions [8,9].

Increased intestinal viscosity, reduced bacterial translocation from the intestine to the liver, and decreased serum levels of an intestinal permeability marker—fluorescein isothiocyanate dextran (FITC-d)—were all seen in recent studies on HS-fed broilers [10]. Mucin-2 (MUC-2) gene expression has also been shown to be positively regulated by HS [11]. According to these findings, HS seems to strengthen the small intestine’s protective mucus layer, which enables the villi to keep their integrity in the face of pathogens, toxins, and dietary alterations.

It appears that, regardless of the mechanism of action of HS, whether as microbiota stabilizers, acidifiers, or mucosal protectors, the end result on the digestive tract is an increased mucosal surface area, increased digestibility, and nutrient absorption, all of which lead to increased productive efficiency [2,4]. Changes in the digestive mucosa of HS-fed broilers have been observed in studies, with increased height, thickness, and surface area of the villi [5,7,12]. These studies were conducted on broilers raised in floor pens, and intestinal samples were collected after 42 days [5,7] and 56 days [12].

There is little data on the effect of HS on mucin production. MUC-2 expression was found to be higher in the cecum mucosa of 17–29-week-old hens in one study [11]. Another study of 1–14-day-old broilers found higher viscosity of intestinal contents after a 24-h fasting period in HS-fed birds [10]. This finding suggests that in HS-fed birds, mucus production is maintained even though chicks had experienced intestinal inflammation as a result of fasting, as evidenced by a decrease in bacterial translocation and FITC-d serum concentration [10]. Previous research has shown that fasting causes villous atrophy in chickens but increases the density of mucin-producing goblet cells (GC) [13,14]. In a previous study, HS-fed broilers offered feed ad libitum and those subjected to alternated restricted feeding for periods of 24 h showed higher carcass yield, lower oocyst excretion, and higher lactic acid bacteria, regardless of the feeding regimen, compared to a negative control group, which indicated that HS could protect against the damaging effects of intestinal inflammation caused by feed restriction for 24 h [15].

In commercial contexts, changes in feed type and formulation are frequently linked to reduced function and inflammatory reactions of the digestive mucosa [16,17]. In the modern fast-growing broiler, as the growth period is progressively shortened and feed efficiency continuously improved, the factors that negatively affect the health and nutrition of the bird should be well identified. Broilers chickens are subjected to frequent dietary changes since diets are typically fed for 5 to 15 days in practice [16]. It is critical to pay attention to the minute changes that occur in the gut because of changes in diet type and formulation, which are frequently overlooked because the damage is subtle and usually characterized by microscopic changes in the mucosal layer. Because there is a vast surface of absorptive epithelial cells beneath the mucosa that is essential for the transport of nutrients into the enterocyte, these minute changes strengthen the efficiency of nutrient assimilation [17]. Furthermore, the mucosa contains components of the immune system as well as mucus-producing goblet cells in the epithelium that protect the host from pathogen damage.

Several mechanisms have been proposed to account for the advantages encountered in broiler chickens supplemented with HS; one of these mechanisms is the ability to create a protective layer over the digestive tract’s epithelial mucosal membrane against the penetration of toxic and other bacterial contaminated substances [10,11]. The inclusion of HS in feeds is likely to reduce subtle, but still harmful, changes to the digestive mucosa while maintaining adequate digestion and immune response, including the maintenance of the protective mucus layer. However, it has not been investigated in HS-fed broilers whether morphological changes in the intestinal mucosa are accompanied by changes in the number of different types of GC in birds reared under no digestive disturbances and under conditions that may cause digestive disorders. Consequently, the goal of this study was to assess the productive performance, morphometric changes, and number of neutral, acid sulfated, and acid non-sulfated GC in the intestinal villi of broiler chickens under steady-state digestive conditions, as well as under sudden dietary changes, with the addition of HS.

## 2. Materials and Methods

The study was conducted at the experimental poultry farm of the National Center for Disciplinary Research in Physiology and Animal Improvement, belonging to the National Institute of Forestry, Agricultural and Livestock Research, located in Ajuchitlan, Queretaro, Mexico. This research was reviewed and approved by the Institutional Subcommittee for Care and Use of Experimental Animals (Protocol Number: SICUAE.DC-2021/2-6) of the National Autonomous University of Mexico.

### 2.1. Animals and Treatments

Ninety Ross 308 male chicks aged 14–38 days were used, each housed individually in battery cages. A complete randomized design with three treatments and 30 replicate cages per treatment was used from 14 to 28 days. A white corn and soybean meal-based basal diet (diet 1) was provided, along with three types of growth promoters: (1) a negative control without antibiotic or coccidiostat (nonGPA), (2) a positive control supplemented with 0.05% bacitracin methylene disalicylate (BMD) and 0.05% salinomycin (GPA), and (3) a negative control supplemented with 0.33% HS (HS). Following previously described procedures, HS were extracted from a worm compost using an alkaline solution [15,18]; a 0.33% dosage was chosen from the results of a previous study [15]. The concentration of humic acids and fulvic acids in the dry product was 43.5% and 29.0%, respectively, for a total of 73.5% HS; the estimated aromaticity was 53.8% [15]. The ash content was 27.5%. The elemental analysis of HS was carried out using energy dispersive X-ray spectroscopy, and the main elemental composition was as follows: O = 45.77%, Na = 27.45%, C = 13.79%, and trace amounts of Si, K, Cl, S, P, and Ca [15]. Previously, the estimated chemical properties and flat structure of humic and fulvic acid molecules with aromaticity were also published [19].

After the broilers were weighed on day 28, the diets were changed, introducing diets A and B, while maintaining the three treatments previously described. Diet A contained white corn, soybean meal, and distillers dried grains with solubles (DDGS), whereas diet B contained half of white corn and half of blue corn, plus soybean meal and DDGS. From 28 to 37 days, diets A and B were fed, and on the morning of day 37, a diet switch was applied by exchanging diet A for broilers that had been fed diet B, and diet B for broilers that had been fed diet A. The diets were introduced on day 28 and exchanged on day 37 to simulate the change of diets commonly carried out at various stages of chicken growth, and thus, to detect subtle changes in histology and the number of GC in the digestive mucosa 24 h after the change was made. It has been reported that the pericarp of blue corn contains a high concentration of phenolic compounds, flavonoid anthocyanins, and natural antioxidants; some of these phenolic compounds, such as tannins, saponines, and toxic alkaloids, may have antinutritional properties [20]. In blue corn, the amount of total phenolic compounds is higher than in white corn due to higher amounts of flavonoid anthocyanins, flavonols, and flavanols; anthocyanins are classified as pigments that confer the tonality to the colored corn [20]. Diet A was expected to have minor changes, while diet B was expected to have major changes on gut morphology after they were suddenly offered. Table 1 shows the composition of the diets. The diet used from 14–28 days and diets A and B were formulated to provide the same amount of nutrients as shown in Table 1. The addition of HS provided 0.04% Na, which was accounted for in the formulation, resulting in a total of 0.21% Na in all diets.

Individual broilers were weighed at the start, as well as on days 28 and 38. Weight gain (WG, g/d) was calculated for the days 14–28, 28–38, and 14–38. During the same time periods, the feed intake (FI, g/d) and the feed conversion ratio (FCR) were calculated. Six broilers from each treatment were killed on day 28, prior to the diet change, and the remaining chicks were vaccinated against Newcastle, Infectious Bronchitis, and Salmonella (AviPro ND-IB2-SE4 CONC). The other six chicks from each treatment were killed on day 29, 24 h after the diet change. On day 38, 24 h after the second diet change, the remaining broilers from each treatment (six per treatment) were killed.

### 2.2. Collection and Processing of the Intestines

Each bird had 5 cm of the middle region of the duodenum, jejunum, and ileum removed; the tissues were rinsed with saline and knotted at the ends. A 10% buffered formalin fixative solution was infiltrated into the intestine lumen with a syringe, and the tissue was placed in vials containing the same fixative solution for 24 h. After 24 h, a 1 cm section was cut around the circumference, the formalin was removed, and the tissue rinsed with distilled water and PBS. Tissues were automatically processed using an automatic tissue processor (Microm, STP 120, Tarragona, Spain), with ethanol used in ascending gradients at 75, 90, and 100% for 6 h, followed by xylene for 6 h, and then embedded in resin.

Each slide had three histological sections from the same treatment bird, for a total of four slides for each bird. Following that, common tissue staining techniques were used: (1) Hematoxylin and eosin staining for villi morphometric analysis (height and thickness), (2) Pass de Schiff staining for quantification of neutral goblet cells, (3) Alcian blue staining at pH 1.0 for quantifying strongly sulfated goblet cells, and (4) Alcian blue staining at pH 2.5 for quantifying acidic GC [21]. Photographs of ten complete and whole villi from each treatment were taken with a brightfield microscope (Axiostar plus, ZEISS, Göttingen, Germany) and a digital color camera (ColorView II SIS^®^). AnalySIS optibasic, SIS^®^ Software was used to analyze the images. The villi’s height was measured from the top of the lamina propria to the tip of the villi. The villi thickness was calculated by measuring and averaging the thickness of three equidistant regions along the villi. The apparent surface area of the villi was calculated using the formula (2π) × (thickness/2) × (height) and expressed in mm^2^ [22,23].

### 2.3. Goblet Cell Count

In the middle region of the jejunum, GCs were identified and quantified. Ten complete and whole villi per animal were photographed using an inverted microscope (Leica DMi8) with a 20× objective and a digital color camera (Leica MC170 HD). The images were analyzed with Leica LAS Interactive Measurement Software. The apparent area was delimited by a length of 250 µm; the lower and upper limits were defined by the natural thickness of the villi; and the surface area was calculated using the irregular rectangle formula. The delimited area was counted for the number of cells reactive to Pass de Schiff, Alcian blue pH 1.0, and Alcian blue pH 2.5 stains. The total number of goblet cells was estimated in the total villi area.

### 2.4. Serum Biochemical Markers

Each bird’s blood was collected in tubes without anticoagulant. The samples were centrifuged at 4 °C for 15 min at 2500 rpm. The serum was separated and stored at −80 °C in microtubes until further use. The serum enzymes were analyzed using spectrophotometry, using a UV/VIS spectrophotometer (GENESYS 10S UV-Vis Thermo scientific) and following the procedures described in the kits (SPINREACT, S.A./S.A.U, Carretera Santa Coloma, España); alanine aminotransferase (ALT; Ref: 41283), aspartate aminotransferase (AST; Ref: 41272), alkaline phosphatase (ALP; Ref: 41242), and urea (Ref: 41043).

### 2.5. Statistical Analysis

The data was subjected to an analysis of variance [24]. A completely randomized design with three treatments was used from 14 to 28 days, with 30 replicate birds per treatment. A completely randomized design with a 3 × 2 factorial arrangement was used from 28–38 and 14–38 days, with three treatments and two diets, and six birds in each cell of the two-way interaction. Since from 28–38 days, no interactions of type of growth promoter and type of diet were observed in the growth performance, data from 14–38 days was analyzed using a complete randomized design with three treatments. To analyze the urea and serum metabolites, as well as the histology and GC number, six birds per treatment were sampled at 28, 29, and 38 days. In addition, the height, thickness, and villi area, as well as the number of neutral, sulfated acid, non-sulfated acid, and total GC cells, were measured in 10 complete villi per bird and there were 60 replicates of each per treatment.

## 3. Results

The results of productive performance are shown in Table 2. From 14–28 and 29–38 days, there was no effect of the treatments on the productive parameters; from 14–38 days, lower FCR was observed in the GPA and HS treatments compared to nonGPA (*p* < 0.05).

The results of serum urea and liver enzyme concentrations at 28, 29, and 38 days are presented in Table 3. On day 28, serum urea, ALT, and AST concentrations did not differ between treatments. FA concentrations were lower with HS than with GPA or nonGPA (*p* < 0.05). On day 29 and day 38, there were no differences in urea, ALT, AST, and FA concentrations between treatments. On day 38, diet A had a lower urea concentration than diet B (*p* < 0.01).

Table 4 shows the villi histology and GC number at day 28. The height of villi in the duodenum, jejunum, and ileum remained consistent across treatments. The villi thickness and area in the duodenum were comparable between treatments; in the jejunum, the villi thickness and area were lower with GPA and HS compared to nonGPA (*p* < 0.01), and in the ileum, the thickness and area were lower with GPA compared to nonGPA and HS (*p* < 0.01). GPA and HS reduced the number of neutral cells, sulfated acids, non-sulfated acids, and total GC per villi (*p* < 0.01).

Table 5 displays the villi histology and GC number at day 29 and 38. There were no differences in villi height between treatments in the duodenum; however, villi thickness was greater (*p* < 0.01) with HS compared to GPA and nonGPA. The thickness and area of villi in the jejunum were greater in broilers fed diet A and supplemented with GPA compared to nonGPA and HS, but similar in diet B (interaction of growth promoter and type of diet, *p* < 0.05; Figure 1a). NonGPA increased the height (*p* < 0.01) and area (*p* < 0.05) of villi in the ileum compared to GPA and HS. Diet A increased the thickness and area of villi in the duodenum (*p* < 0.01) compared to diet B. Diet A increased the height of the villi in the jejunum compared to diet B (*p* < 0.05).

Diet A reduced the height, thickness, and area of villi in the ileum compared to diet B (*p* < 0.05). The number of neutral, acidic, non-sulfated cells, and total GC was higher with GPA (*p* < 0.01), intermediate with HS, and lower with nonGPA, and was higher (*p* < 0.01) with diet A compared to diet B. The number of sulfated acid cells was higher in broilers fed diet A with GPA compared to nonGPA and HS; there were no differences among treatments in diet B (interaction of growth promoter and type of diet, *p* < 0.01; Figure 1b).

Table 5 shows the villi histology and goblet cell number at day 38. The height and area of the villi in the duodenum were lower with GPA compared to HS and nonGPA (*p* < 0.05); the thickness was comparable between treatments. There were no differences in villi height between treatments in the jejunum. The thickness and area of the villi were greater with GPA and nonGPA compared to HS with diet A and lower with GPA compared to nonGPA and HS with diet B (interaction of growth promoter and type of diet, *p* < 0.01; Figure 2a). In the ileum, villi height was similar between treatments with diet A, and with diet B; villi height was lower with GPA and HS compared to nonGPA (interaction of growth promoter and type of diet, *p* < 0.05); villi thickness and area were greater (*p* < 0.05) with nonGPA compared to GPA and HS.

The number of neutral cells was higher with GPA and nonGPA compared to HS with diet A, but lower with GPA compared to nonGPA and HS with diet B (interaction of growth promoter and type of diet, *p* < 0.01; Figure 2b). The number of sulfated acid cells was higher with GPA, intermediate with nonGPA, and lower with HS in diet A, but lower with GPA, intermediate with HS, and higher in diet B (interaction of growth promoter and type of diet, *p* < 0.01; Figure 3a). The number of non-sulfated acid cells and total GC were higher with GPA and nonGPA compared to HS with diet A, and lower with GPA compared to nonGPA and HS with diet B (interaction of growth promoter and type of diet, *p* < 0.01; Figure 3b).

## 4. Discussion

When compared to non-AGP-fed broilers, AGP-fed broilers had a lower FCR from 14–38 days and smaller villi in the duodenum, jejunum, and ileum on day 28. Lower FCR [25,26], lower GIT weight per unit BW [27], and lower villi height [26,28] in BMD-fed broilers are consistent with these findings. GPA has several advantages, including reduced bacterial growth, decreased local inflammation due to pathogen control, and increased intestine size [29,30]. Furthermore, on days 29 and 38, broilers given AGP had a reduced villi area in the duodenum and ileum. The lack of an effect of AGP on villi area reduction in the jejunum on days 29 and 38 may be attributed to the abrupt dietary change on days 28 and 37, respectively. When compared to the area on day 28, the villi area in the jejunum of GPA-fed broilers increased by 14% and decreased by 19% on day 29 with diet A and B, respectively. The area of the villi was 30% smaller in broilers fed diet B than in broilers fed diet A. This suggests that the presence of GPA in diet B did not protect broilers from mucosal atrophy. The antimicrobial effect of BMD appears to have been preserved in broilers fed diet A with DDGS, but this effect was likely negated by the combination of DDGS and blue corn in diet B. A transient dysbiosis that resulted in villous atrophy could have caused this. At day 38, the area of the villi in the jejunum was lower in AGP-fed broilers fed diet B, but was similar to nonAGP-fed broilers fed diet A. This discrepancy is difficult to understand.

Higher FCR and villi area in the duodenum (non-significant), jejunum, and ileum at day 28, the ileum at day 29, and the duodenum, jejunum, and ileum at day 38 were observed in non-AGP-fed broilers. These findings support the hypothesis that pathogenic bacteria in the mucosa increase GIT thickness and nutrient requirements in nonAGP-fed broilers by increasing mucosal turnover [26,29]. The area of the villi in the duodenum of nonAGP-fed broilers was similar to that of AGP-fed broilers on day 29, but it was larger than that of nonAGP-fed broilers on day 28. However, compared to day 28, the villus area in the jejunum was reduced by 38% in broilers fed diets A and B. This suggests that the sudden presence of DDGS and the combination of DDGS and blue corn caused villi atrophy, which was linked to the lack of an antimicrobial effect; the magnitude of the reduction in villus area was comparable between nonAGP-fed broilers receiving diet A and B.

When HS-fed broilers were compared to AGP-fed broilers at day 28, they had similar FCR from 14–38 days and a larger area of villi in the jejunum, but a smaller area of villi in the duodenum (non-significant) and the ileum. Previous research found that HS-added broilers had lower FCR [6,31,32], even when supplemented with HS extracted from worm compost [18,33]. HS has previously been discussed in terms of its effects on poultry growth and health [1,2,34]. Other studies have found that HS-fed broilers [5,7,12] and rats [35] have increased villi height, thickness, and surface area. These findings support the larger area of villi in the duodenum and ileum but contradict the smaller area of villi in the jejunum in AGP-fed broilers. The results at day 28 indicate that in broilers fed HS for 14 days in a low-stress environment, the area of the villi differs in each segment of the small intestine. The villi area in the jejunum behaves similarly to that of AGP-fed broilers, whereas the villi area in the duodenum and ileum behaves similarly to that of non-AGP-fed broilers. The reason for these differences is unknown.

At day 29, 24 h after the introduction of diets A and B, the villi area of HS-fed broilers was comparable to that of AGP-fed broilers in the duodenum and jejunum with diet B, and in the ilium with diet A and B; in the jejunum of birds receiving diet A, the villi area of HS-fed broilers was lower than that of AGP-fed broilers. Furthermore, these findings contradict previous research that found trophic effects on the intestinal mucosa of HS-fed broilers [5,7,12]. At day 28, the villi area in the jejunum decreased by 12% and 22% with diet A and B, respectively, when compared to the villi area in HS-fed broilers. The proposed mucosal protective effect of HS was completely reversed 24 h after the switch of diets, with diet B causing more villi atrophy. This also implies that, in the face of a sudden dietary change, and despite the low addition of DDGS in diet A, HS did not prevent mucosal damage as well as GPA did.

The area of the villi in the duodenum of HS-fed broilers was larger at days 28, 29, and 38 when compared to AGP-fed broilers; no clear trends were observed in the jejunum; and higher villi area at day 28, but similar area at days 29 and 38, when compared to AGP-fed broilers were shown in the ileum. The duodenum findings are more consistent with previous publications in HS-fed broilers than the jejunum and ileum findings. Previous studies were conducted in floor pens, and intestinal samples were collected after broilers were fed HS for 42 days [5,7] and 56 days [12], whereas in the current study, birds were penned individually and fed a diet containing HS for 28 days.

The neutral, sulfated acid, non-sulfated acid, and total GC followed a pattern similar to that seen in the villi area of the jejunum on day 28. In germ-free hens, the absence of bacteria in the GIT resulted in a decrease in the amount and density of neutral and acidic GC, the absence of sialylated GC, and a decrease in the expression of mucin 2 mRNA compared to chickens exposed to a single bacteria population or cecal bacterial contents [36]. Because of the antimicrobial activity of BMD and salinomycin, broilers supplemented with GPA in the current study imitated the findings reported in germ-free chickens. However, the lower GC number in HS-fed broilers contrasts with higher viscosity in the small intestine [10] and higher MUC-2 gene expression [11] previously reported. In these studies, broilers were reared in groups in floor-pens in both reports, and viscosity in the ileum [10] and the MUC-2 gene in the ceca were measured [11]. Opposite to this, in our study, broilers were reared in individual cages and GC were counted in the jejunum.

On day 29, the number of neutral, sulfated acid, non-sulfated acid, and total GC in the jejunum decreased by an average of 27, 31, 34, and 33% when compared to day 28. The proportional drop in GC, on the other hand, varied between treatments. The proportional reduction of neutral cells in AGP-, nonAGP-, and HS-fed broilers was 3, 47, and 24%, respectively, in sulfated cells, 19, 38, and 36%, and in non-sulfated cells, 13, 55, and 26%. In AGP-, nonAGP-, and HS-fed broilers, the average drop was 12, 51, and 29%, respectively. These findings show that AGP-fed broilers had the smallest drop in GC, while non-AGP-fed broilers had the greatest drop, with HS-fed broilers showing an intermediate drop, particularly in the counts of neutral and non-sulfated cells. Increased GC density but a decreased number of enterocytes per villi and decreased villi surface area in chicks subjected to feed restriction after hatch [13,37,38], as well as in 39-day-old broilers after 24 h of feed withdrawal [14], have been reported. The increased GC density in the villi was thought to be caused by a decrease in villi surface area or a heightened host defense mechanism triggered by the feed restriction [38]. The increased density of GC reported in feed restricted birds is consistent with the higher number of neutral, non-sulfated, and total GC in AGP- and HS-fed broilers after 24 h of dietary change compared to nonAGP-fed broilers. The number of neutral, sulfated acid, non-sulfated acid, and total GC in the jejunum decreased by an average of 6, 16, 15, and 12% at day 38 when compared to day 28. It has been proposed that the GC population in the small intestine may mature at 3 weeks of age due to a slower rate of cell differentiation and migration on day 21 compared to days 7 and 14 [39,40]. Furthermore, it has been claimed that broiler GC density is constant between the third and sixth weeks of life [13,41]. The decrease in GC in 38-day-old broilers compared to 28-day-old broilers in the current study could be attributed to the exchange of diets on day 37.

The number of neutral, sulfated acid, non-sulfated acid, and total GC at day 38 followed the same pattern as the area of the villi of the jejunum at day 38, in that diet A was lower in HS-fed broilers and diet B was lower in AGP-fed broilers. Surprisingly, the total GC number in AGP-, nonAGP-, and HS-fed broilers decreased by 8, 21, and 5%, respectively, when compared to day 28 values. This implies that HS, like AGP, had similar effect on the number of GC following the second diet change. This result supports the theory that HS strengthens the intestinal mucosal barrier against bacteria or toxins released by bacteria in the lumen [8,9].

Summarizing the results regarding the growth promoters, HS-fed broilers had a similar FCR to AGP-fed broilers from 14–38 days. From 14–28 days, the villi area was greater in the duodenum and ileum and similar in the jejunum when compared to AGP-fed broilers. The villi area of HS-fed broilers was comparable to that of AGP-fed broilers in the duodenum and jejunum with diet B and in the ileum with diet A and B on day 29, 24 h after the abrupt change in diets. The villi area was higher in the duodenum but lower in the ileum at day 38, 24 h after the diet change, while it was lower and higher in the jejunum with diet A and B, respectively, compared to AGP-fed broilers. These results indicate that there was no clear trend in the effect of HS on the morphometry of villi in broilers raised under stable digestive conditions and following abrupt dietary changes. The amount of neutral, sulfated acid, non-sulfated acid, and total GC in the jejunum of HS-fed broilers was comparable to that of AGP-fed broilers at day 28. The GC number fell after the dietary change on day 29, but the drop was intermediate in HS-fed broilers compared to AGP and nonAGP-fed broilers; this response was stronger in broilers fed diet A. The GC number dropped after the diet change at day 37 as well; however, this decrease was lower in HS- and AGP-fed than in non-AGP-fed broilers. These findings suggest that under stable digestive conditions and after experiencing sudden dietary changes, the number of GC in the jejunum of HS-fed behaves similarly to that of AGP-fed broilers.

The abrupt introduction of diets A and B on day 28 was expected to cause changes in gut morphology by exposing the broilers to a moderate amount of fiber (diet A) and fiber plus diverse phenolic compounds (diet B). It has been previously indicated that in commercial contexts, changes in feed type and formulation are frequently linked to reduced function and inflammatory reactions of the digestive mucosa [16,17]. In broilers fed diets A and B, the area of the villi increased by 33% and 15% in the duodenum, 14% and 26% in the jejunum, and 5% and 21% in the ileum, respectively, at day 29. The villi in the duodenum were longer and wider than on day 28, the villi in the jejunum were shorter and thinner, and the villi in the ileum were shorter but wider. The residence time and feed component digestion processes that occur in each segment of the intestine most likely influenced the reduced villi area in the jejunum. Retention times in the duodenum, jejunum, and ileum are reported to be 5, 40–60, and 90–120 min, respectively [42,43]. Because the digestion process is very limited before the jejunum and the majority of the diet components arrive intact, the dietary change had a more damaging effect on the mucosal surface in this segment. The sudden addition of fiber in diet A and fiber plus phenolic compounds in diet B may have increased mucosal shedding, resulting in a decrease in the height, thickness, and area of the jejunum villi. It is possible that a transient dysbiosis with concurrent bacterial damage to the digestive mucosa occurred [13]. The greater villi area in the jejunum of diet A-fed broilers may also account for the higher number of neutral, acidic non-sulfated cells and total GC compared to diet B-fed broilers. Blue corn phenolic compounds were most likely degraded by microbiota enzymes and disappeared before reaching the ileum [44]. This may account for the increased villi area in the ileum, instead of atrophy being observed as in the jejunum.

There were no significant effects of diet type on the villi area at day 38. Regardless of diet, the villi area increased by 35% in the duodenum, 4% in the jejunum, and 6% in the ileum when compared to day 28. There were no discernible effects of diet type on GC number. Regardless of diet, the number of neutral, sulfated, non-sulfated, and total GC cells in the jejunum decreased by 6, 16, 15, and 12%, respectively, compared to day 28. Because the area of villi in the jejunum was only reduced by 4%, the greater reduction in the number of sulfated and non-sulfated cells in this segment is unexplained. However, this could be due to a mucosal adaptation process after broilers were fed diet A and B from 29–37 days.

It is known that the duodenal mucosa is exposed to dietary components for a brief period of time, while the majority of feed components are absorbed and digested by the end of the jejunum. The pancreas secretes several digestive enzymes, and the gallbladder secretes bile salts to the lumen of the jejunum, and many membrane-associated enzymes and protein transporters are found in the microvilli of the cell membrane of the luminal side of the enterocytes [42,43]. The ileum appears to be the primary intestinal section in charge of mineral and water absorption. However, on days 29 and 38, the jejunal villus area decreased by 20% and 4%, respectively, while the duodenal villus area increased by 24% and 36% and the ileum increased by 13% and 4%. If these segments have limited digestion and absorption of nutrients, the significance of maintaining and extending mucosal surface area in the duodenum and ileum compared to the jejunum is unknown. The presence of hypertrophied intestinal absorptive epithelial cells on the apical surface of the ileum’s villi, as well as a larger absorptive surface of the duodenum’s villi, has previously been linked to improved performance [45,46]. In the present research, a larger villus area in the duodenum and ileum may have resulted in greater demand for nutrients to support an increased cellular turnover rate, whereas a smaller villus area in the jejunum may have saved nutrients for use in other body functions. This subject requires more explanation.

According to previous research [47], longer intestinal villi may indicate a superior capacity for feed absorption in the intestine, and this has been associated with active cell mitosis, which increases the villi’s capacity to absorb nutrients [48,49]. In the present study, although villus height in the duodenum, jejunum, and ileum did not differ between treatments at day 28, there was greater villus area in the jejunum (nonAGP) and ileum (nonAGP and HS) due to increased villus thickness. On day 29, the height of the villi in the duodenum was comparable between treatments, but diet A had more villi area due to the thicker villi. At day 38, there were no differences in the height of the villi of the jejunum and ileum; however, differences in villi thickness resulted in greater villi area in some treatments. These results indicate that in several instances, the villi were shorter but wider, which leads to a greater villi area. The findings also imply that the capacity of the intestine to increase the area of digestion and absorption is not solely dependent on the height of the villi or the relationship between the height of the villi and the depth of the crypts, because the villi thickness should also be considered if used in the various formulas proposed to calculate the area of the villi.

## 5. Conclusions

HS-fed broilers had a similar FCR to AGP-fed broilers from 14–38 days. The results indicate that there were no clear trends in the effect of HS on the morphometry of villi in broilers raised under stable digestive conditions (day 28) and following abrupt dietary changes (days 29 and 38). It was also found that under stable digestive conditions, the number of GC in the jejunum of HS-fed broilers behaves similarly to that of AGP-fed broilers. After experiencing a sudden dietary change, the GC number fell on day 29, but the drop was intermediate in HS-fed broilers compared to AGP and nonAGP-fed broilers; at day 38, the drop in GC number was lower in HS- and AGP-fed compared to non-AGP-fed broilers. HS appears to strengthen the mucus protection in the epithelium of the intestine.

## Figures and Tables

**Figure 1 animals-13-00212-f001:**
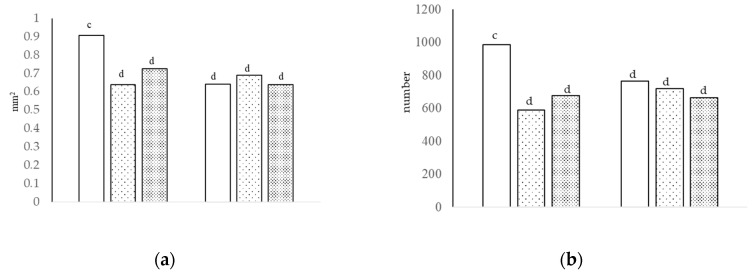
The villi area [(**a**), interaction of growth promoter and type of diet, ^c,d^
*p* < 0.05; SEM = 0.045] and number of sulfated acid cells [(**b**), interaction of growth promoter and type of diet, ^c,d^
*p* < 0.01; SEM = 51.823] in the jejunum at day 29.

**Figure 2 animals-13-00212-f002:**
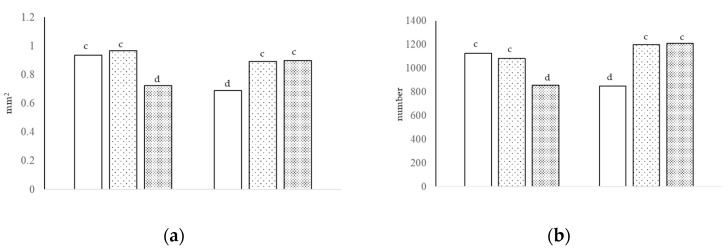
The villi area [(**a**), interaction of growth promoter and type of diet, ^c,d^
*p* < 0.01; SEM = 0.057] and number of neutral cells [(**b**), interaction of growth promoter and type of diet, ^c,d^
*p* < 0.01; SEM = 81.758] in the jejunum at day 38.

**Figure 3 animals-13-00212-f003:**
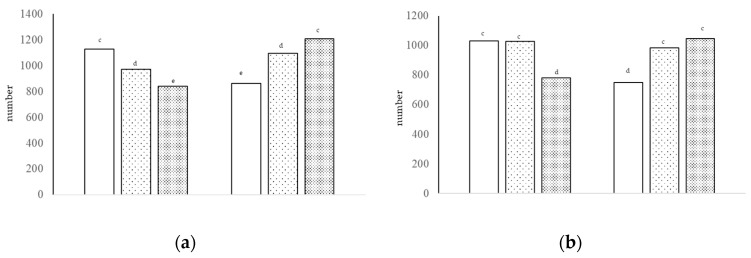
Number of sulfated acid cells [(**a**), interaction of growth promoter and type of diet, ^c–e^
*p* < 0.01; SEM = 76.253] and number of non-sulfated acid cells [(**b**), interaction of growth promoter and type of diet, ^c,d^
*p* < 0.01; SEM = 71.298] in the jejunum at day 38.

**Table 1 animals-13-00212-t001:** Ingredient composition of the basal diets.

	14–28 Days	28–38 Days	
Diet ^1^		A	B
Ingredients, kg			
White corn	63.98	63.51	31.76
Blue corn	0.00	0.00	31.76
Soybean paste	29.81	25.10	25.10
DDGS	0.00	3.50	3.50
Vegetable oil	2.33	3.30	3.30
Calcium orthophosphate	1.23	1.48	1.48
Calcium carbonate	1.45	1.45	1.45
Vitamins and minerals ^2^	0.50	0.50	0.50
Salt	0.33	0.30	0.30
Sodium bicarbonate	0.20	0.20	0.20
Methionine	0.00	0.20	0.20
Lysine	0.00	0.24	0.24
Threonine	0.00	0.05	0.05
Choline chloride	0.07	0.07	0.07
BMD ^3^	0.05	0.05	0.05
Coccidiostat	0.05	0.05	0.05
Calculated nutrient content			
ME, Kcal/kg	3100	3100	3100
Crude protein, %	19.41	19.97	20.30
Digestible Lys, %	1.00	1.00	1.00
Digestible Met, %	0.38	0.38	0.38
Digestible Thr, %	0.65	0.65	0.65
Ca, %	0.95	0.95	0.95
Available P, %	0.40	0.40	0.40
Na, %	0.21	0.21	0.21
Cl, %	0.20	0.20	0.20

^1^ DDGS = distiller’s dried grains with solubles ^2^ Each kg provided: 6500 IU Vit A; 2000 IU Vit D3; 15 IU Vit E; 1.5 mg Vit K; 1.5 mg thiamine; 5 mg riboflavin; 35 mg niacin; 3.5 mg pyridoxine; 10 mg pantothenic acid; 1500 mg choline; 0.6 mg folic acid; 0.15 mg biotin; 0.15 mg Vit B12; 100.0 mg Mn; 100 mg Zn; 50 mg Fe; 10 mg Cu; 1.0 mg I. ^3^ BMD = bacitracin methylene disalicylate.

**Table 2 animals-13-00212-t002:** Effect of the type of growth promoter on the productive performance from 14 to 38 days ^a,b^.

	GPA	nonGPA	HS	SEM ^c^	*p*-Value
Body weight					
14 days	435.08	441.51	428.15	10.167	0.46
28 days	1267.23	1184.01	1216.01	38.782	0.34
38 days	2146.58	2105.75	2155.17	30.355	0.36
Productive variables from 14 to 28 days					
Feed intake, g/d	93.89	93.49	93.24	0.549	0.70
Weight gain, g/d	60.46	59.03	59.85	2.260	0.90
Feed conversion	1.56	1.61	1.57	0.064	0.84
Productive variables from 14 to 38 days					
Feed intake, g/d	132.65	133.61	133.68	1.762	0.90
Weight gain, g/d	71.31	69.34	71.96	1.153	0.18
Feed conversion	1.85 ^d^	1.93 ^e^	1.86 ^d^	0.023	0.05

^a^ Data are means of six replications (birds) per treatment. ^b^ GPA = growth promoter antibiotic; nonAGP = free of growth promoter antibiotic; HS= humic substances; ^c^ Standard error of the mean. ^d,e^ Values within rows with different superscripts differ significantly, *p* < 0.05.

**Table 3 animals-13-00212-t003:** Effect of the type of growth promoter on the serum urea and liver enzyme concentrations at 28, 29, and 38 days ^a,b^.

	GPA	nonGPA	HS	SEM ^c^	*p*-Value
Serum urea and liver enzyme concentrations at day 28					
Urea, mmol/L	0.29	0.45	0.28	0.080	0.27
ALT, U/L	7.44	10.31	10.65	1.366	0.22
AST, U/L	95.08	114.87	89.49	13.231	0.39
FA, U/L	1455.94 ^d^	1508.10 ^d^	596.38 ^e^	254.930	0.05
Serum urea and liver enzyme concentrations at day 29					
Urea, mmol/L	0.27	0.30	0.24	0.038	0.55
ALT, U/L	6.68	5.86	4.79	0.688	0.17
AST, U/L	90.93	91.86	84.90	6.345	0.70
FA, U/L	1666.23	1109.12	1406.35	204.245	0.17
Serum urea and liver enzyme concentrations at day 38					
Urea, mmol/L	0.35	0.25	0.22	0.041	0.10
ALT, U/L	2.04	1.77	2.45	0.426	0.55
AST, U/L	56.13	59.14	52.78	3.607	0.77
FA, U/L	1246.85	1700.46	1533.08	139.816	0.10

^a^ Data are means of six replications (birds) per treatment. ^b^ GPA = growth promoter antibiotic; nonAGP = free of growth promoter antibiotic; HS = humic substances; ALT = alanine aminotransferase; AST = aspartate aminotransferase; FA = alkaline phosphatase; ^c^ Standard error of the mean. ^d,e^ Values within rows with different superscripts differ significantly, *p* < 0.05.

**Table 4 animals-13-00212-t004:** Effect of the type of growth promoter on the villus histology and goblet cell number at day 28 ^a,b^.

Item	GPA ^a^	nonGPA	HS	SEM^c^	*p*-Value
Duodenum					
Height, µm	1385	1431	1399	48.513	0.79
Thickness, µm	214	239	236	9.190	0.11
Area, mm^2^	0.937	1.096	1.065	0.061	0.16
Jejunum					
Height, µm	1227	1265	1306	43.760	0.45
Thickness, µm	208 ^d^	257 ^e^	197 ^d^	9.551	0.01
Area, mm^2^	0.794 ^d^	1.038 ^e^	0.822 ^d^	0.050	0.01
Ileum					
Height, µm	682	686	731	28.770	0.41
Thickness, µm	169 ^d^	197 ^e^	192 ^e^	8.426	0.01
Area, mm^2^	0.363 ^d^	0.432 ^e^	0.433 ^e^	0.025	0.01
Number of goblet cells in the villi of the jejunum					
Neutral	959 ^d^	1337^e^	1064^d^	80.599	0.01
Sulfated acid	1079 ^d^	1352^e^	1044^d^	80.574	0.01
Non-sulfated acid	982 ^d^	1342^e^	996^d^	69.875	0.01
Total	3020 ^d^	4030^e^	3104^d^	213.055	0.01

^a^ Data are means of six replications (birds) per treatment. ^b^ GPA = growth promoter antibiotic; nonAGP = free of growth promoter antibiotic; HS = humic substances; ^c^ Standard error of the mean. ^d,e^ Values within rows with different superscripts differ significantly, *p* < 0.01.

**Table 5 animals-13-00212-t005:** Effect of the interaction of the type of growth promoter and type of diet on the villus histology and goblet cell number at day 29 and 38 ^a,b^.

	Diet A ^a^			Diet B				*p*-Value		
Item	GPA	nonGPA	HS	GPA	nonGPA	HS	SEM ^c^	TD	TGP	TD * TGP
Villus histology and goblet cell number at day 29										
Duodenum										
Height, µm	1518	1630	1564	1451	1622	1491	101.683	0.15	0.62	0.39
Thickness, µm	254	244	314	254	222	260	11.132	0.01	0.01	0.05
Area, mm^2^	1.252	1.296	1.59	1.152	1.181	1.225	0.108	0.01	0.28	0.49
Jejunum										
Height, µm	1186	1116	1166	1117	1102	1039	38.995	0.05	0.40	0.39
Thickness, µm	232	178	191	181	197	191	8.35	0.11	0.05	0.01
Area, mm^2^	0.908	0.640	0.726	0.643	0.691	0.638	0.045	0.01	0.05	0.01
Ileum										
Height, µm	609	716	590	660	687	645	21.908	0.14	0.01	0.10
Thickness, µm	220	199	215	219	255	235	11.239	0.01	0.83	0.05
Area, mm^2^	0.426	0.457	0.4	0.46	0.558	0.468	0.029	0.01	0.05	0.43
Number of goblet cells in the villi of the jejunum										
Neutral	964	758	897	903	661	711	56.897	0.01	0.01	0.53
Sulfated acid	987	590	678	765	719	663	51.823	0.40	0.01	0.01
Non-sulfated acid	929	632	888	779	571	592	59.467	0.01	0.01	0.12
Total	2880	1971	2462	2446	1952	1967	156.488	0.01	0.01	0.26
Villus histology and goblet cell number at day 38										
Duodenum										
Height, µm	1580	1667	1650	1401	1637	1670	66.603	0.24	0.05	0.30
Thickness, µm	250	288	283	267	255	282	13.086	0.60	0.18	0.16
Area, mm^2^	1.241	1.557	1.496	1.212	1.387	1.511	0.095	0.43	0.01	0.60
Jejunum										
Height, µm	1217	1178	1046	1148	1143	1156	42.554	0.94	0.14	0.08
Thickness, µm	243	248	212	186	239	243	10.173	0.16	0.05	0.01
Area, mm^2^	0.935	0.966	0.723	0.689	0.892	0.898	0.057	0.30	0.06	0.01
Ileum										
Height, µm	671	661	708	638	724	623	30.09	0.40	0.35	0.05
Thickness, µm	190	205	181	188	239	177	7.193	0.14	0.01	0.06
Area, mm^2^	0.409	0.451	0.413	0.382	0.542	0.355	0.012	0.91	0.01	0.06
Number of goblet cells in the villi of the jejunum										
Neutral	1125	1083	855	850	1197	1208	81.758	0.34	0.16	0.01
Sulfated acid	1129	971	839	864	1094	955	76.253	0.94	0.3	0.01
Non-sulfated acid	1029	1026	782	750	982	1047	71.598	0.74	0.24	0.01
Total	3284	3080	2476	2464	3273	3250	210.174	0.78	0.24	0.01

^a^ Data are means of six replications (birds) per treatment. ^b^ GPA= growth promoter antibiotic; nonAGP = free of growth promoter antibiotic; HS = humic substances; TD = type of diet; TGP = type of growth promoter; ^c^ Standard error of the mean.

## Data Availability

The data that support the findings of this study are available from the corresponding author upon reasonable request.

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
