# Peer review of "Effect of the Addition of Humic Substances on Morphometric Analysis and Number of Goblet Cells in the Intestinal Mucosa of Broiler Chickens"

_animals, 2023, doi:10.3390/ani13020212_

Round 1
Reviewer 1 Report
1. Line 61, “…broilers. (FITC-d) [10].” What means?
2. Line 81, “…broiler chickens subjected to low-stress rearing condition…”, how did the authors construct low stress conditions?
3. What is the reason for the dies A and B changed for two times between 28 and 38 days of age? Please explain in the manuscript.
4. From the point of view of the experimental design, the author sampled 24 hours after changing the diet, and whether such a short period of time has an effect on the intestinal tract of broilers is questionable.
5. why did the authors choose 0.33% HS?
6. line 119, please define the abbreviation of FCR.
7. line 162, “goblet cells” changed to “GC”.
8. table 2, did diets (A and B) have an effect on broiler growth performance, serum urea and liver enzyme concentrations?
9. Table 4, Please give the data for each of the three treatments under the respective diet conditions. The same as table 5.
10. Figure 1, we can not see the interaction of growth promoter and type of diet. What “*”stands for? the rest of the 5 bars are marked with "**". To whom are they compared? The same as figure 2 and figure 3. Therefore, please present the data in this article in a more understandable way.
Author Response
- Line 61, “…broilers. (FITC-d) [10].” What means?
Response: the description of FITC-d was made
- Line 81, “…broiler chickens subjected to low-stress rearing condition…”, how did the authors construct low stress conditions?
Response: to avoid confusion, the term was changed to mean “steady-state digestive conditions”
- What is the reason for the dies A and B changed for two times between 28 and 38 days of age? Please explain in the manuscript.
Response: the rationale for changing diets is presented in the Introduction and in the Materials and methods
- From the point of view of the experimental design, the author sampled 24 hours after changing the diet, and whether such a short period of time has an effect on the intestinal tract of broilers is questionable.
Response: The rationale for changing diets is presented in the Introduction and in Materials and methods. In brief, the objective of sampling 24 hours after the diet change was to detect subtle changes in the intestinal epithelium, that may impair the digestive process, but are overlooked when sampling is done several days after the change. A summary of these changes in the mucosa is as follows: A dramatic atrophy was found in the jejunum, whereas in the duodenum and ileum, hypertrophic responses were observed at days 29 and 38 compared to day 28. On day 29 the area of the villi in the jejunum dropped by 18% and the total GC dropped by 31%, compared to values on day 28. At day 38 the area of the villi in the jejunum dropped by 4% and the total GC dropped by 11%, respectively, compared to values on day 28. Since the jejunum is the main segment of nutrient digestion and absorption, the severe atrophy observed 24 hours after the diet change may be associated with a transient reduction of growth or impaired feed efficiency. Probably the lower FCR in the non-AGP treatment was due to greater villous atrophy at 29 and 38 days.
- why did the authors choose 0.33% HS?
Response: the 0.33% HS was chosen from the results of previous research: Domínguez-Negrete, A.; Gómez-Rosales, S.; Angeles, M.d.L.; López-Hernández, L.H.; Reis de Souza, T.C.; Lator-re-Cárdenas, J.D.; Téllez-Isaias, G. Addition of Different Levels of Humic Substances Extracted from Worm Compost in Broiler Feeds. Animals 2021, 11, 3199. https://doi.org/10.3390/ani11113199
- line 119, please define the abbreviation of FCR.
Response: the definition of FCR was made
- line 162, “goblet cells” changed to “GC”.
Response: the correction was made
- table 2, did diets (A and B) have an effect on broiler growth performance, serum urea and liver enzyme concentrations?
Response: Diets A and B did not have essential effects on serum urea and liver enzymes; in Results on day 38, only Diet A had a lower urea concentration than diet B (p < 0.01).
- Table 4, Please give the data for each of the three treatments under the respective diet conditions. The same as table 5.
Response: Tables 4 and 5 were modified to show the results of the interaction of growth promoter and type of diet; tables were combined.
- Figure 1, we can not see the interaction of growth promoter and type of diet.What “*”stands for? the rest of the 5 bars are marked with "**". To whom are they compared? The same as figure 2 and figure 3. Therefore, please present the data in this article in a more understandable way.
Response: The given interactions were described in the Results; in addition, the symbols * were changed for appropriate superscript letters
Reviewer 2 Report
The study evaluated the performance enhancing potential of humic substances in broiler nutrition and provided insights on its mechanism of action at the gut level. The manuscript could benefit from providing more information below-
Line 102- Information on what informed the 0.33% HS dosage of supplementation could be provided.
Lines 113-11- More rationale for diet switching again on day 37 may be provided.
Table 2 and all other result tables- It is always a good idea to include the specific P values on the result table and let the readers decide for themselves, if significant or not.
Minor comments
Line 23- “was study” should be “was studied”
Lines 33-34 and 100-103- Diets could be renumbered with the negative control (no GPA- no HS) coming first. It aids easier understanding.
Line 60, 61- (FITC-d) was meant to be immediately after intestinal permeability marker and not in lines 61
Line 61- muc2 is mucin 2 and not “mucine 2”
Line 175- information on the statistical software utilized for analysis could be provided.
Author Response
Thank you very much for your recommendations
The study evaluated the performance enhancing potential of humic substances in broiler nutrition and provided insights on its mechanism of action at the gut level. The manuscript could benefit from providing more information below-
Line 102- Information on what informed the 0.33% HS dosage of supplementation could be provided.
Response: the 0.33% HS was chosen from the results of previous research: Domínguez-Negrete, A.; Gómez-Rosales, S.; Angeles, M.d.L.; López-Hernández, L.H.; Reis de Souza, T.C.; Lator-re-Cárdenas, J.D.; Téllez-Isaias, G. Addition of Different Levels of Humic Substances Extracted from Worm Compost in Broiler Feeds. Animals 2021, 11, 3199. https://doi.org/10.3390/ani11113199 . This information was entered in Materials and methods
Lines 113-11- More rationale for diet switching again on day 37 may be provided.
Response: the rationale was described in the Introduction and in the Materials and methods. In brief, the switching of diets was done to simulate the change of diets in broilers at different stages of growth
Table 2 and all other result tables- It is always a good idea to include the specific P values on the result table and let the readers decide for themselves, if significant or not.
Response: the p < values were entered in all tables
Minor comments
Line 23- “was study” should be “was studied”
Response: the correction was made
Lines 33-34 and 100-103- Diets could be renumbered with the negative control (no GPA- no HS) coming first. It aids easier understanding.
Response: The correction was made.
Line 60, 61- (FITC-d) was meant to be immediately after intestinal permeability marker and not in lines 61
Response: The correction was made.
Line 61- muc2 is mucin 2 and not “mucine 2”
Response: the correction was made
Line 175- information on the statistical software utilized for analysis could be provided.
Response: the information of the statistical package was introduced in the statistical analysis
Reviewer 3 Report
Line 97 - the numbers should be written rather than words ("ninety")
Line 101 - all abbreviations must be explained when 1st time appear (i.e. BMD)
Line 111 - the schema of the experiment is very complicated. Why diets were switched? It is not motivated in the introduction
Line 166 - how many blood samples were taken?
Table 2 - the production effects should be presented separately from biochemical serum indices. It is difficult to accept that in case of FA analysis results differences between twice bigger values (or more) are not statistically significant. It should be verified again
Table 3 - instead of "a, b" superscripts the number of samples may be given in brackets
All tables - instead of very complicated marking using superscripts the p-value of particular factors as well as their interaction influence should be added
The transposition of table 4 and 5 is recommended
Table 5 - there is doubts in case of goblet cells statistical verification. They shouldn't be analysed usin non-parametric test? the "total" means "sum" or mean value of all samples? How many samples were analysed each time?
Figures - Why differences of 3 values are marked using asterisks? It is acceptable in case of 2 values, then single asterisk means differences at p<0.05, double at 0.01, etc. What was compared here?
Line 293-294 - how it is possible in 24 hrs?
Author Response
Thank you very much for your recommendations
Line 97 - the numbers should be written rather than words ("ninety")
Response: the correction was made
Line 101 - all abbreviations must be explained when 1st time appear (i.e. BMD)
Response: the correction was made
Line 111 - the schema of the experiment is very complicated. Why diets were switched? It is not motivated in the introduction
Response: The rationale for changing diets is presented in the Introduction and in Materials and methods. In brief, the objective of sampling 24 hours after the diet change was to detect subtle changes in the intestinal epithelium, that may impair the digestive process, but are overlooked when sampling is done several days after the change. A summary of these changes in the mucosa is as follows: A dramatic atrophy was found in the jejunum, whereas in the duodenum and ileum, hypertrophic responses were observed on days 29 and 38 compared to day 28. On day 29 the area of the villi in the jejunum dropped by 18% and the total GC dropped by 31%, compared to values on day 28. At day 38 the area of the villi in the jejunum dropped by 4% and the total GC dropped by 11%, respectively, compared to values on day 28. Since the jejunum is the main segment of nutrient digestion and absorption, the severe atrophy observed 24 hours after the diet change may be associated with a transient reduction of growth or impaired feed efficiency. Probably the lower FCR in the non-AGP treatment was due to greater villous atrophy at 29 and 38 days.
Line 166 - how many blood samples were taken?
Response: six samples of blood were taken per treatment on each day of sampling. This point was clarified in the Statistical analysis
Table 2 - the production effects should be presented separately from biochemical serum indices. It is difficult to accept that in the case of FA analysis results differences between twice bigger values (or more) are not statistically significant. It should be verified again
Response: Table 2 was split. I apologize for omitting the appropriate superscripts in the Table to show the differences at day 28 in FA concentrations; however, these differences had been already described in the Results. The results are shown in Table 3.
Table 3 - instead of "a, b" superscripts the number of samples may be given in brackets
Response: The actual statistical values of each variable response were introduced upon request of Reviewer 1.
All tables - instead of very complicated marking using superscripts the p-value of particular factors as well as their interaction influence should be added
Response: The actual statistical values of each variable response were introduced upon request of Reviewer 1. The means and p< values of the interaction of the type of growth promoter and type of diet was given in Table 5.
The transposition of table 4 and 5 is recommended
Response: Tables 4 and 5 were merged
Table 5 - there is doubts in case of goblet cells statistical verification. They shouldn't be analysed usin non-parametric test? the "total" means "sum" or mean value of all samples? How many samples were analysed each time?
Response: All GC numbers are discontinuous variables and are normally distributed. The mean values shown in the tables were rounded, but the SEM shows the using up to three decimal place differences after the period.
Figures - Why differences of 3 values are marked using asterisks? It is acceptable in case of 2 values, then single asterisk means differences at p<0.05, double at 0.01, etc. What was compared here?
Response: The symbols * were changed for appropriate superscript letters; the description of differences shown in each figure was given in Results.
Line 293-294 - how it is possible in 24 hrs?
Response: The rationales for these changes were described in the Introduction at the request of reviewer 1; in the present study, diets A and B were formulated to exacerbate and detect the alterations caused by the diet change in the period immediately after the change, since these alterations are not detected when sampling is performed several days after the change.
Round 2
Reviewer 1 Report
The author's revision is satisfactory, so this manuscript can be accepted in the present form.
Author Response
Thank you
Reviewer 3 Report
If the exact p-value is given in the table should be simple written as "p-value" not "p<". If it is p-value of interaction of 2 factors it should be also mentioned in the table head. And still marking of statistical differences is difficult to read.
Author Response
Response: the “p-value” is shown in all tables and all results tables now have a more descriptive title.
Thank you very much for the time you have spent on reviewing our manuscript. Your comments are very valuable and helpful for revising our paper and guiding our research.